# Environmental and Anthropogenic Influence on the Core Beneficial Honeybee Gut Microbiota—A Short Communication from Bulgaria

Svetoslav G. Dimov 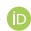

Department of Genetics, Faculty of Biology, Sofia University, "St. Kliment Ohridski", 1164 Sofia, Bulgaria; svetoslav@biofac.uni-sofia.bg; Tel.: +359-2-8167342

**Abstract:** Bees' and beehives' health are strongly influenced by the honeybees' gut microbiota which in turn is strongly dependent on many different factors, including environmental factors as well as anthropogenic pressure. In this study, in four locations in Bulgaria differing strongly in environmental conditions and anthropogenic pressure, an assessment was made using several obligatory core symbiont species and genera for reference, such as *Lactobacillus* sp*., Bifidobacterium* sp*., Snodgrassiella alvi, Gilliamella apicola, Frishella perrara,* and *Commensalibacter* sp., as well as an observation of the overall number of species. A snapshot of the relative abundances of the total number of species and the core species was made using a next-generation sequencing (NGS)-based metagenomic approach using the Illumina 2 × 250 bp paired-end platform. It was found that the two forms of anthropogenic pressure, the agricultural and the urban/industrial, have distinct effects, affecting different core genera and species. It was also demonstrated that both types of anthropogenic pressure cause a reduction in the overall number of bacterial species.

**Keywords:** honeybee gut microbiota; metagenomics; anthropogenic pressure

## 1. Introduction

It is difficult to estimate the exact economic value of honeybee pollination in worldwide agriculture, with assessments reaching several billion USD in developed countries, as well as up to a trillion USD worldwide. A good picture can be drawn from the work of Hein [1]. However, these calculations do not include the economic impact of honeybee pollination on wild ecosystems. It was well documented that honeybees' microbiotas can change with time and age, as well as according to their functions in hives and within the different parts of the gastrointestinal tract [2–4]. However, certain core species are globally omnipresent because of their role in modulating bees' fitness and health [3,5,6]. In this process, a major role is played by several ubiquitous beneficial symbionts such as several members of the genera *Lactobacillus* and *Bifidobacterium*, as well as *Snodgrassella alvi*, *Gilliamella apicola*, *Frischella perrara*, and *Commensalibacter* sp. [7]. Because beekeeping is not only a traditional livelihood in Bulgaria but also a major pollination source for its mostly agriculture-based economy, understanding the abiotic and biotic factors behind hives' health and wellness is very important. In this research, the main goal was to implement a V3-V4 16S-based metagenomics analysis to evaluate the content of these species and genera in the honeybee *Apis mellifera macedonica* at the end of June 2020 during the most active foraging season in four locations in Bulgaria differing strongly in climatic conditions, landscape, and anthropogenic pressure. Because anthropogenic pressure is manifested in the form of changes in both biotic and abiotic environmental factors, and is expressed in two forms, the urban/industrial and the agricultural, the focus of this study was on the reflection of both forms in the honeybees' gut microbiotas.

## 2. Results

The quality control statistics indexes of the generated NGS data, the taxonomic annotation analyses, and the alpha-diversity indexes (Shannon, Simpson, Chao1, ACE, and Good's coverage) are summarized in Table 1.

**Table 1.** Summarization of the results from the metagenomics analyses.

| | | Kalina | Sofia | Momchilovtzi | Dushantzi |
|---|---|---|---|---|---|
| | Number of raw paired-end reads | 139,384 | 138,342 | 106,761 | 130,041 |
| | Number of raw tags | 107,840 | 114,233 | 88,950 | 96,505 |
| | Number of clean tags | 105,840 | 112,012 | 86,962 | 95,339 |
| | Number of effective tags | 92,726 | 100,564 | 70,750 | 92,140 |
| Quality control statistics of the NGS data | Total number of bases of effective tags | 38,396,863 | 42,476,920 | 29,967,266 | 37,496,294 |
| | Average length of effective tags | 414 | 422 | 424 | 407 |
| | Q20 [1] | 98.17 | 98.16 | 98.07 | 98.14 |
| | Q30 [1] | 94.20 | 94.15 | 93.93 | 94.07 |
| | Percentage of GC content | 52.78 | 51.85 | 51.54 | 54.99 |
| | Percentage of effective tags in raw paired-end reads | 66.53 | 72.69 | 66.27 | 70.85 |
| | Total tags | 92,726 | 100,564 | 70,750 | 184,280 |
| Tags data and tags annotation data | Taxon tags | 91,561 | 99,301 | 69,413 | 183,708 |
| | Unclassified tags | 0 | 2 | 0 | 8 |
| | Unique tags | 1165 | 1261 | 1337 | 564 |
| | Observed species | 162 | 84 | 231 | 322 |
| | Shannon | 3.042 | 3.165 | 4.012 | 2.585 |
| Alpha diversity indexes | Simpson | 0.712 | 0.819 | 0.875 | 0.728 |
| | Chao1 | 163.500 | 85.000 | 231.000 | 322.000 |
| | ACE | 163.921 | 85.427 | 231.000 | 322.000 |
| | Good's coverage | 1.000 | 1.000 | 1.000 | 1.000 |

[1] Q20 and Q30 are the percentages of bases whose quality value in effective tags is greater than 20 (sequencing error rate is less than 1%) and 30 (sequencing error rate is less than 0.1%). NGS: next-generation sequencing.

The level of community diversity was evaluated via calculation of the Shannon and Simpson indices. The community richness was estimated via calculation of the Chao1 and ACE indices. The Good's coverage index was used to assess the sequencing depth.

The results from the complete annotation of the operational taxonomic units (OTUs) can be retrieved from Supplementary File S1.

The main genera distributions among the different samples were compared via Krona charts (Figure 1).

The percentages of the beneficial eubacterial ubiquitous gut symbionts in different locations were calculated based on the Krona chart results, and are listed in Table 2.

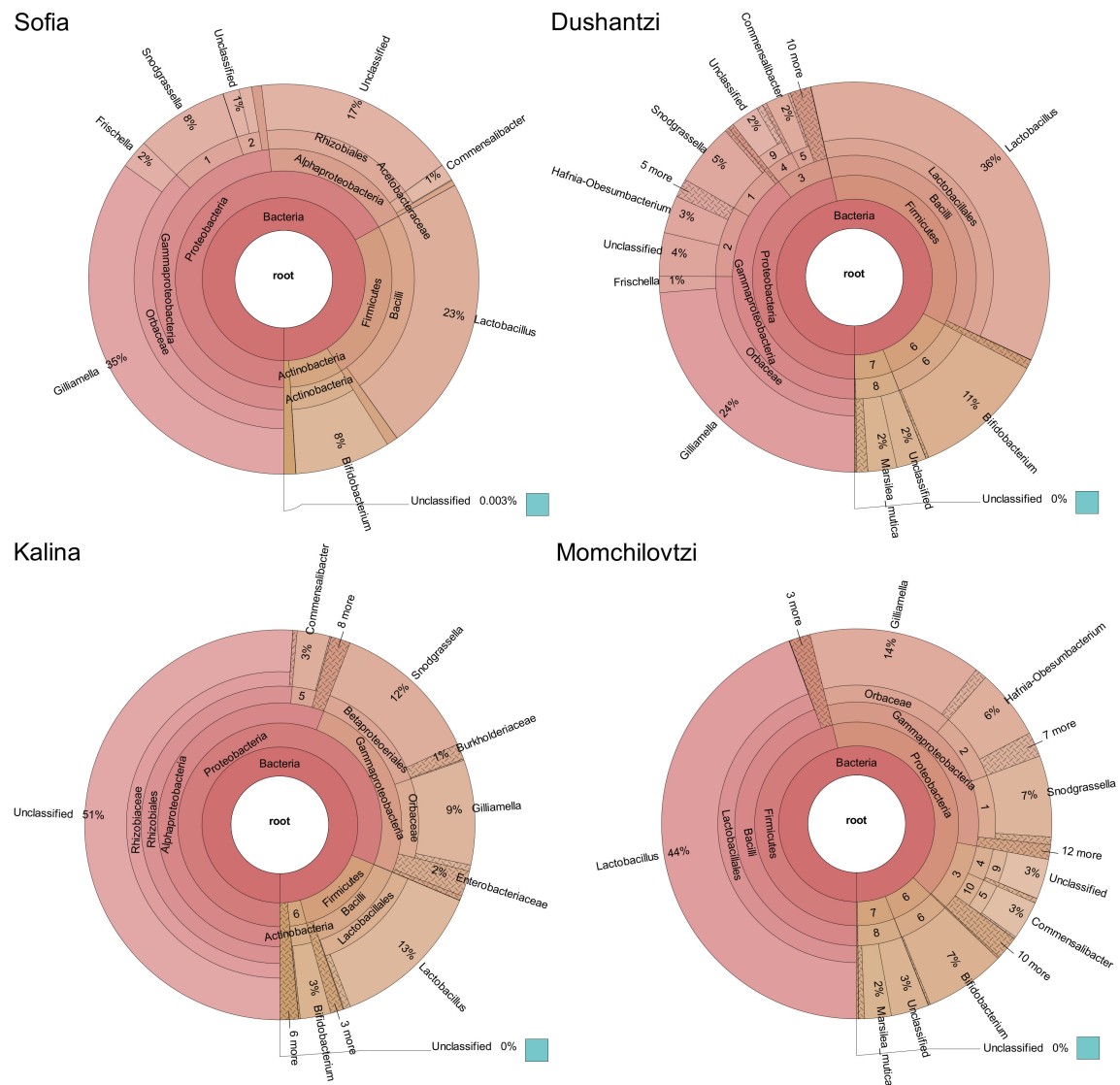

**Figure 1.** Krona charts of the bacterial genera distributions among the samples from the four locations. Phyla listed by numbers: 1—Betaproteobacteriales; 2—Enterobacteriaceae; 3—Alphaproteobacteria; 4—Rhizobiales; 5—Acetobacteriaceae; 6—Actinobacteria; 7—Oxyphotobacteria; 8—Choloroplast; 9—Rhizobiaceae; 10—Acetobacteriales.

**Table 2.** Percentages of the beneficial eubacterial ubiquitous honeybee gut symbionts in the four locations in Bulgaria [1].

| | *Lactobacillus* sp. | *Bifidobacterium* sp. | *Snodgrassiella alvi* | *Gilliamella apicola* | *Frischella perrara* | *Commensalibacter* sp. | Total % |
|---|---|---|---|---|---|---|---|
| Sofia | 23 | 8 | 8 | 35 | 2 | 1 | 77 |
| Dushantzi | 36 | 11 | 5 | 24 | 1 | 2 | 79 |
| Momchilovtzi | 44 | 7 | 7 | 14 | 1 | 3 | 76 |
| Kalina | 13 | 3 | 12 | 9 | 1 | 3 | 41 |
| Average % | 29 | 7 | 8 | 21 | 1 | 2 | 68 |

[1] The percentages are rounded to an integer.

## 3. Discussion

Until now, very few studies have addressed the correlation of honeybees' core species microbiota composition and environmental factors. One exception and a good example is the study of Jones et al. [8] who proved that there is a correlation between the landscape and

the diversity and composition of the overall gut microbiota of the honeybee, as well as with the relative abundances of the core species. Still, in the study of Jones et al., only two types of landscapes were included—both agricultural. In this research, four types of landscapes were considered, including different degrees of industrial/urban and agricultural pressure.

The quality control statistics indexes of the generated NGS data (Table 1) guaranteed the correct interpretation of the further taxonomic annotation analyses, especially by the high values of the Q20 and Q30 indexes, as well as by the total number of bases of the effective tags, the average length of the effective tags, and the percentages of the effective tags in the raw paired-end reads. At this early stage of analysis, differences in the GC content suggested differences in the gut microbiota composition at the four locations.

This study, as a preliminary one, includes only 12 beehives in four apiaries, and thus a correct statistical analysis is not possible. Still, the differences in the overall microbiota composition were further confirmed by the taxonomic annotation analyses which revealed interesting tendencies (Table 1). In all cases, the percentages of the taxon tags were very high, while the quantity of the unclassified tags was insignificant. However, the numbers of observed species varied significantly, and they did not correlate with the numbers of taxon tags, nor with the numbers of unique tags. The highest numbers of species were observed in Momchilovtzi and Dushatzi, 231 and 322 species respectively, where the anthropogenic influence is less pronounced. The smallest numbers of species were observed in Sofia (84) and Kalina (162), where the natural landscape is almost absent, and the anthropogenic pressure is very high. The negative impact of the artificial ecosystems and the anthropogenic pressure is even more pronounced if the numbers of the observed species are compared with those of the taxon tags, displaying a reduction in the composition and evenness of the microbiotas. The values of the alpha diversity indexes (Table 1), especially those of the Chao1 and ACE indexes coinciding with the number of observed species and resulting in a Good's coverage value of one, imply that these analyses should be considered as correct and informative [9] (the complete OTUs annotation results can be found in Supplementary File S1).

However, the main focus of this study was on the beneficial ubiquitous gut symbionts, and thus the content of these species and genera were investigated separately using the results of the Krona charts (Figure 1) [10].

Different authors report up to ten core genera and species of honeybee gut symbionts; however, only *Lactobacilli*, *Bifidobacteria*, *Snodgrassiella alvi*, and *Gilliamella apicola* are ubiquitous [11]. In the present study, in addition to these, the potentially probiotic *Frischella perrara* and *Commensalibacter* sp. from the non-obligatory core species were also observed at the four locations, and were therefore included within the analyses (Table 2).

The core species contribute to the beehive's welfare by two major mechanisms: food digestion and immunity. *Bifidobacteria* and *Lactobacilli*, together with *G. apicola* and *Commensalibacter* sp., participate in pollen degradation and sugar breakdown [12,13]. The primary role of *S. alvi*, in a partnership with *G. apicola*, is the formation of a protective biofilm [14]. In contrast, *F. perrara* plays a defensive role by stimulating the immune response [15].

A high content of lactobacilli and *Commensalibacter* sp. has been reported for thriving beehives, while a high content of bifidobacteria and *G. apicola* has been observed in non-thriving hives [13]. In this regard, it was not surprising to find the highest percentages of *Lactobacillus* sp. in Momchilovtzi and Dushantzi, where the anthropogenic pressure is relatively low or almost non-existent, while the lowest percentages of the genus were observed in Kalina, a location with a very strong anthropogenic pressure of an agricultural nature, and Sofia, where the anthropogenic pressure is also very strong but of an urban/industrial type.

Large differences in the relative content of *G. apicola* were also observed. The highest percentage was observed in Sofia, but surprisingly the lowest was observed in Kalina; thus, it can be speculated that only the urban pressure affects the content of the species, while the agricultural pressure does not. However, despite its immunity-promoting role, it was not possible to affirm a correlation between the *G. apicola* content and the presence or

absence of pathogens because pathogen-specific PCR tests were not performed, and thus the possibility of including infected bees in the DNA pool could not be excluded.

The differences in the content of the other core species found in this study were less obvious due to the similar percentages found at the different locations. However, for *Bifidobacterium* sp., the agricultural anthropogenic pressure led to a reduction in the proportion, while the urban/industrial pressure did not. The agricultural pressure, and to a lesser extent the urban/industrial pressure, were reflected in the *S. alvi* content by raising it, most likely due to its role in the stress-protective response. It is likely that the anthropogenic pressure negatively impacts the content of *Commensalibacter* sp., since the lowest percentage was observed in Sofia. No tendencies were observed for *F. perrara*.

A significant difference in the overall core microbiota content was observed in Kalina, where the total percentage of the six species and genera was only 41%, while for the other three locations it varied between 76% and 79%. This undoubtedly means that the agricultural pressure had a clear negative impact on the percentages of the beneficial bacteria, while the urban/industrial pressure did not. This could possibly be explained by the vast surrounding fields planted with sunflowers as a monoculture, which thus significantly reduce the diversity of bees' diet.

In Dushantzi and Momchilovtzi, where the highest numbers of species were observed, as well the highest percentages of the beneficial core microbiota, the food sources are much more diverse. It is generally accepted that higher numbers of species make the gut microbiota functionally more efficient and stable. The lowest number of species was observed in Sofia; however, in the capital city, a high percentage of the ubiquitous honeybee gut symbionts was observed, thus confirming once again the adaptive capacity of the honeybee to an urban environment.

Finally, even though the number of samples in this study did not allow statistical analysis, the observed tendencies are logical and clear, and reveal that V3-V4 NGS-based metagenomics represents a powerful and time-efficient tool for studying insects' microbiotas.

## 4. Materials and Methods

### 4.1. Sampling Locations and Their Environmental Characteristics

The four apiaries locations are shown in Figure 2.

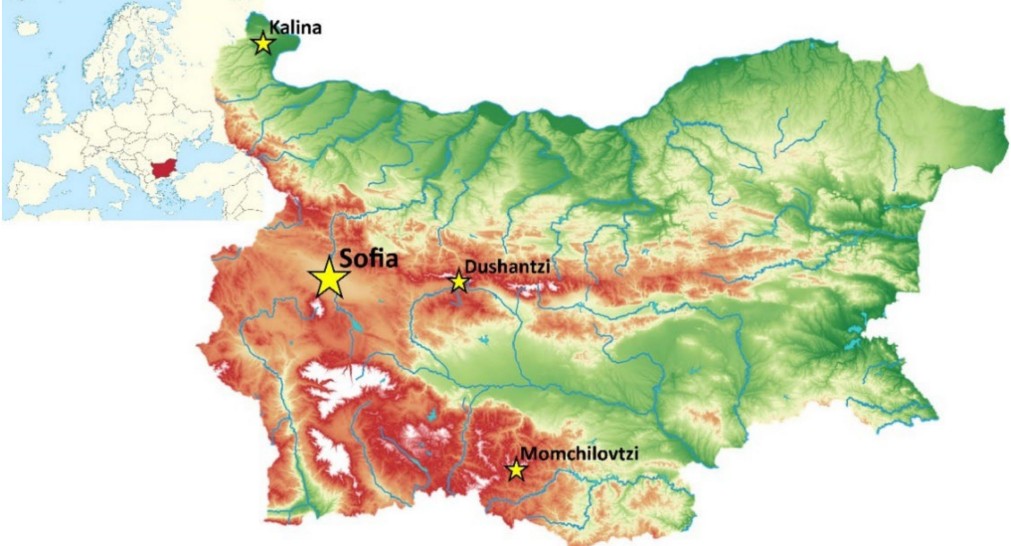

**Figure 2.** Map of the locations of the apiaries from which honeybees have been provided [16].

### 4.1.1. Sofia

Geographic coordinates: 42.744263 N, 23.264150 E; elevation: 550 m; the capital city of Bulgaria; population over 1,600,000 inhabitants; continental climate; extremely intensive road traffic and many industrial enterprises, including thermal power plants; scarce agricultural activities. The anthropogenic pressure is estimated to be very high because of urbanization and the presence of many industrial enterprises.

### 4.1.2. Dushantzi

Geographic coordinates: 42.698124 N, 24.262389 E; elevation: 720 m; pre-mountain rural area in the mid-western part of the country; continental climate; residents—700; developed but not very intensive agriculture; the presence of many preserved natural areas and a big dam; no industrial enterprises. The anthropogenic pressure is estimated to be moderate.

### 4.1.3. Kalina

Geographic coordinates: 44.068890 N, 22.767405 E; elevation: 105 m; located in the most north-western region of the country in the large Danube plane; continental climate; residents—about 40; very intensive agriculture; the presence of a small dam used for irrigation; no industrial enterprises. The anthropogenic pressure is estimated to be high because of intensive agriculture.

### 4.1.4. Momchilovtzi

Geographic coordinates: 41.657372 N, 24.773179 E; elevation 1180 m; located in the southern part of Bulgaria in the middle of the Rhodopes mountain range; continental climate but with a strong influence from the Aegean Sea which is at a distance of approximately 60 km; residents—1050; sheep breeding and some other animal farming in the high-mountain pastures as principal agricultural activities; no industrial enterprises. The anthropogenic pressure is estimated to be very low.

### *4.2. DNA-Based Techniques*

Four pools made from the guts of three different bees from three different hives in each apiary were made. DNA was extracted from each of the pools with "Quick-DNA™ Fecal/Soil Microbe Microprep Kit" (Zymo Research, Irvine, CA, USA, Cat. No. D6012). The DNA concentrations were measured by Quantus™ Fluorimeter (Promega, Madison, WI, USA), while the integrity of the DNA was analyzed electrophoretically on 1% agarose gels in a TBE buffer system.

### *4.3. Next-Generation Sequencing*

The DNA samples were shipped in dry ice to the Novogene Company Ltd. (Cambridge, UK). The sequencing was performed on the Illumina HiSeq $2 \times 250$ bp paired-end reads platform with 30 k tags per sample, using as an amplicon target the V3-V4 region of the 16S rRNA genes.

### *4.4. Bioinformatics Analyses*

The bioinformatics processing of the obtained data was as previously described [16], and the results were organized as interactive files. The raw data were uploaded to the NCBI (BioProject: PRJNA771483, accession numbers: SRX12629443 (Sofia), SRX12629442 (Dushantzi), SRX12629441 (Momchilovtzi), SRX12629440 (Kalina)). For the OTU analyses, the Mothur software was used in conjunction with the SSUrRNA database of the SILVA database for species annotation at each taxonomic rank [17,18].

## 5. Conclusions

In this preliminary study, which is also a pioneering one for Bulgaria, it was discovered that the environmental conditions and the anthropogenic pressure each impact the content

of the core bacterial species and genera of the honeybee gut microbiota. The two types of anthropogenic pressures, the agricultural and the urban/industrial, have different effects, reflected in the content of different species and genera. However, the agricultural pressure was revealed to be stronger, leading to a net reduction in the content of the beneficial core genera and species. On the other hand, the urban/industrial pressure resulted in a clear reduction in the total number of eubacterial species.

**Supplementary Materials:** The following supporting information can be downloaded at: https://www.mdpi.com/article/10.3390/bacteria1020008/s1, Supplementary File S1: Complete list of the OTUs annotation results.

**Funding:** This research was funded by the Bulgarian National Research Fund, grant number КП-06-H26/8 from 17.12.2018.

**Institutional Review Board Statement:** Not applicable.

**Informed Consent Statement:** Not applicable.

**Data Availability Statement:** Raw Illumina sequencing data is publicly available at the National Center for Biotechnology Information under BioProject # PRJNA771483.

**Acknowledgments:** The author would like also to express his gratitude to the following beekeepers for their cooperation and for providing them the bees: Atanas Atanasov from Kalina, Andrei Andreev from Momchilovtzi, Martin Dimitrov from Dushantzi and Dimitar Dimitrov from Sofia.

**Conflicts of Interest:** The author declares no conflict of interest. The funder had no role in the design of the study; the collection, analyses, or interpretation of data; in the writing of the manuscript, or in the decision to publish the results.

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
