# Peer review of "Environmental and Anthropogenic Influence on the Core Beneficial Honeybee Gut Microbiota—A Short Communication from Bulgaria"

_2674-1334, doi:10.3390/bacteria1020008_

Round 1
Reviewer 1 Report
Within the present manuscript, preliminary results about a detailed investigation of the composition of bee gut microbiota against the background of different sampling regions is presented. The data show that the anthropogenic pressure negatively influences the diversity of bee's gut microbiota. Even though this is a Short Communication, some additional information would help a better understanding and interpretation of the manuscript:
Lines 35-39: The goal of the research activities is stated. Please also shortly provide the motivation (why has this topic been investigated) to perform the research.
Line 44/Table 1: Different alpha diversity indexes are given. Please shortly introduce the concept of the different indexes (e.g. based on reference 9), since the broader audience might not be used to these indexes.
Line 50: Please provide the full name for OTU.
Discussion: Since the work group already investigated beebread (reference 16), it would be interesting to include a discussion about potential correlations between beebread and bee gut microbiota.
Author Response
- “Lines 35-39: The goal of the research activities is stated. Please also shortly provide the motivation (why has this topic been investigated) to perform the research.”
I am very grateful for this suggestion! The motivation will be added as suggested.
- “Line 44/Table 1: Different alpha diversity indexes are given. Please shortly introduce the concept of the different indexes (e.g. based on reference 9), since the broader audience might not be used to these indexes.”
I would like to thank the reviewer also for this suggestion – the meanings of the indices were included within a short paragraph in the Materials and methods section.
- “Line 50: Please provide the full name for OTU.”
The sentence was corrected as suggested.
- “Discussion: Since the work group already investigated beebread (reference 16), it would be interesting to include a discussion about potential correlations between beebread and bee gut microbiota.”
I completely agree with the reviewer that such correlation would be very interesting, however, the presence of the gut core symbiont species within the beebread is very low (usually below 0.1%), so the extrapolated correlations will be speculative at a high degree. Nonetheless, in future studies, such correlations could be formulated but in a different experimental design, for example, based on real-time PCR which will allow more exact enumerations of less presented species.
Reviewer 2 Report
The manuscript ID: bacteria-1649369 by Dimov, entitled “Environmental and anthropogenic influence on the core beneficial honeybee gut microbiota - a short communication from Bulgaria” aim to give insight into bacterial diversity in honeybees Apis mellifera macedonica in four locations in Bulgaria which differ in climate, landscape and anthropogenic conditions.
General remarks:
The proposed short communication concerns the very important subject of pollination and the possible consequences of anthropogenic pollution of the environment. The advantage of the manuscript is that samples were taken from different sites and DNA was isolated from bee’s guts and not whole bees as some studies do. For future research reconsider if three bees from three different hives in each apiary are sufficient to make reliable conclusions. Pooled DNA from 10 bees per hive may be more reasonable, however, I do not know how laborious is the process.
As for bioinformatic analyses I recommend using Qiime 2 pipeline and SILVA database for ASV (amplicon sequence variant) analyses. In my opinion, is the most updated and reliable methodology. In the 4.4 section add short information on what software/database was used.
In my opinion, presentation of the results with Krona charts is a bad approach and uch better would be bar charts.
In discussion or future research, it would be interesting (if possible) to relate the results to e.g. chemical pollution, moise pollution of the environment or resulting bee maturation and mortality.
Also, I am curious if the bees in different locations fed on similar plants - which is rather unlikely. Food may have a high impact on the gut microbiome of various living creatures, in the case of bees exposure to propolis is also important.
I am somewhat surprised that the agricultural environment leads to a net reduction of the content of the beneficial bacteria. However, it is in general accepted that a more diverse microbial community is functionally more efficient and stable. In my opinion, this should be emphasized in the case of agricultural samples.
Author Response
- “The proposed short communication concerns the very important subject of pollination and the possible consequences of anthropogenic pollution of the environment. The advantage of the manuscript is that samples were taken from different sites and DNA was isolated from bee’s guts and not whole bees as some studies do. For future research reconsider if three bees from three different hives in each apiary are sufficient to make reliable conclusions. Pooled DNA from 10 bees per hive may be more reasonable, however, I do not know how laborious is the process.”
I would like to thank the reviewer for the high esteem of this research! Considering the number of the pooled guts we choose to work with three bees’ guts because of the limitations of the sample’s volume of the kit used for DNA extraction (one bee gut weights about 70-80 mg while the maximum sample volume of the kit is 250 mg). Furthermore, the bees from the different hives in the same apiary are more or less in contact with each other visiting the same foraging fields and stealing from each other.
- “As for bioinformatic analyses I recommend using Qiime 2 pipeline and SILVA database for ASV (amplicon sequence variant) analyses. In my opinion, is the most updated and reliable methodology. In the 4.4 section add short information on what software/database was used.”
Indeed, differences in the relative abundances could be observed by comparing the Mothur and Qiime software, however, they concern the less presented microorganisms which were not subject of the present study focused on the core gut symbionts. A good comparison of both software against the SILVA database is the work of López-García et al., 2018 (ttps://doi.org/10.3389/fmicb.2018.03010). A small section indicating the OTUs analyses were performed with the Mothur against the SILVA database was included in the text as suggested.
- “In my opinion, presentation of the results with Krona charts is a bad approach and uch better would be bar charts”.
I agree that bar charts are much more graphically perceptive, however, they would be based only on the 6 bacterial core species if compared to the Krona charts where other genera are also indicated, as well the percentages of the unclassified OTUs. To overcome the perceptional drawback of the Krona charts we included Table 2.
- “In discussion or future research, it would be interesting (if possible) to relate the results to e.g. chemical pollution, moise pollution of the environment or resulting bee maturation and mortality.”
I would like to thank the reviewer for this excellent suggestion! Eventually, in a future project, I will also recruit specialists which can measure and characterize the different types of pollution. Another idea we have in mind is to simulate different types of pollution in our newly established experimental beekeeping station.
- “Also, I am curious if the bees in different locations fed on similar plants - which is rather unlikely. Food may have a high impact on the gut microbiome of various living creatures, in the case of bees exposure to propolis is also important.”
The reviewer is absolutely right – the four locations are completely different: Kalina village is surrounded by enormous monoculture fields (sunflower), Momchilovtzi is most close to the natural environment with sheep breeding in natural high-mountain pastures, Dushantzi where different cultures are cultivated with livestock breeding, and the capital city Sofia with 100% urban and industrial (within the district) environment. That is why we expected that the different environments should reflect on the bees’ microbiotas but we choose to put the focus on the core obligatory symbionts on which the health and wellness of the bees depend on.
- “I am somewhat surprised that the agricultural environment leads to a net reduction of the content of the beneficial bacteria. However, it is in general accepted that a more diverse microbial community is functionally more efficient and stable. In my opinion, this should be emphasized in the case of agricultural samples.”
I am very grateful for this suggestion, and an emphasis was put within the “Discussion” section. A net reduction was observed only in Kalina village where are present vast fields with sunflower as monoculture, while in Dushantzi and Momchilovtzi this was not the case.